# Scratch Testing of AlSi12/SiC_p_ Composite Layer with High Share of Reinforcing Phase Formed in the Centrifugal Casting Process

**DOI:** 10.3390/ma13071685

**Published:** 2020-04-04

**Authors:** Anna Janina Dolata, Marek Mróz, Maciej Dyzia, Magdalena Jacek-Burek

**Affiliations:** 1Faculty of Materials Engineering, Department of Advanced Materials and Technologies, Silesian University of Technology, Krasińskiego 8, 40-019 Katowice, Poland; maciej.dyzia@polsl.pl; 2Department of Foundry and Welding, Rzeszow University of Technology, Al. Powstańców Warszawy 12, 35-959 Rzeszów, Poland; mfmroz@prz.edu.pl (M.M.); m.jacek@prz.edu.pl (M.J.-B.)

**Keywords:** aluminum metal matrix composites, silicon carbide reinforcement, centrifugal casting, composite layer, scratch testing

## Abstract

The scratch test enables assessing the susceptibility of a material to the development of scratches and, being in some ways a measure of its abrasion resistance, allows extended knowledge in the field of material application usability, especially its machining capabilities. The aim of the study was to assess the resistance of a centrifugally formed AlSi12/SiC_p_ composite layer with a high share of reinforcing phase (V_p_ > 40%) to scratching with a diamond indenter. The microstructure and effect of the load applied to the diamond indenter on the scratch depth and susceptibility of the composite layer to the nucleation and propagation of cracks in hard and brittle SiC particles were analyzed. A simple model of SiC_p_ cracking depending on their size, shape (geometry), and orientation in relation to the direction of scratching has been proposed.

## 1. Introduction

The intensive development of mechanical engineering industries, especially the commonly observed trend to reduce the weight and minimize not only the manufacturing and operating cost but also the overall environmental effect of the final product, inclines manufacturers to seek after new solutions in scope of the used engineering materials. An answer to the demand of modern industry are composite materials based on polymer [1], ceramic [2], or metallic [3] matrices.

Metal matrix composites (MMCs) are numbered among new materials showing great application potential [3,4,5]. From among MMCs, those perceived as particularly attractive and most intensively developed are particulate reinforced metal matrix composites (PRMMCs) based on lightweight aluminum alloys (aluminum matrix composites, AlMCs) reinforced with hard ceramic phase, especially with particulate silicon carbide (SiC_p_). A number of authors [6,7,8] pointed out that compared to the matrix material, Al/SiC_p_ composites are characterized with higher strength and rigidity, also at elevated temperature, better stability of the thermal coefficient of expansion, and increased hardness and resistance to abrasive wear. Due to a number of favorable properties, the composites can successfully substitute monolithic materials such as aluminum alloys or iron alloys in applications involving operation in difficult and complex conditions, especially in tribological couples of various types. Therefore, the primary areas of application for AlMCs are components designed for aircraft, automobile, and machine-building industries such as combustion engine cylinder liners, pistons, and brake drums and discs [9,10,11].

Despite the many advantageous service properties of Al/SiC_p_ composites, an important limitation preventing the materials from wider use for machine components and parts is their machinability. According to the literature of the subject [12] and the results of the present authors’ own studies [13], the introduction of hard ceramic particles into the plastic matrix of aluminum alloy makes Al/SiCp composites exceptionally difficult to machine. Basic problems connected with the application of material removal processes to Al/SiC_p_ composites include inadequate surface quality after machining and the excessive wear of tool cutting edges due to the non-homogeneity of the machined material. For this reason, the process of forming finished composite articles requires the employment of special machining tools, such as e.g., diamond tool tips, or alternative advanced material removal processes such as Electro Discharge Machining (EDM), Laser Beam Machining (LBM), or Abrasive Water Jet (AWJ) cutting [14,15,16], which significantly increases the cost per piece of the final product. 

Another economically viable solution consists of using centrifugal casting molds in order to obtain precision castings with a high accuracy of dimensions and form called also the near net shape castings. An additional merit of casting processes in which the effect of centrifugal force is used consists of the possibility of controlling the distribution of the reinforcement in the course of casting solidification [17,18,19,20,21] and thus to improve the service properties of selected superficial heavily loaded areas only. The effectiveness of the process of centrifugal casting of pistons with AlSi18CuMgNi reinforced locally with SiC_p_ was described by Huang and collaborators [22]. Studies carried out by Rajan and Pai [21] confirm that the centrifugal casting technique can be successfully used to manufacture aluminum gear wheels, cylinder linings, and brake discs with hybrid reinforcement in the form of silicon carbide and graphite (GR_p_) particles. The quoted authors have shown that in the case of cylinder linings, it was possible to obtain composite castings with functionally graded structures. 

Similar results were obtained and reported in the present authors’ papers, such as e.g., [23,24,25]. It was shown that the use of hybrid suspensions of Al/SiC_p_ + GC_p_ type enables obtaining cylindrical centrifugal castings with two functional composite layers of different thickness and favorable tribological properties correlated with the type, share (V_p_), and diameter (D_p_) of reinforcing particles. Proper choice of the material and technological parameters of the process (including, for instance, the composite suspension casting temperature, the mold temperature and speed of rotation, and the casting solidification time) enables forming a distinct outer composite layer, the structure and properties of which are determined by the presence of SiC_p_ securing the proper hardness, surface load capacity, and wear resistance. On the other hand, the inner layer of casting is the area in which vitreous carbon particles (GC_p_) surrounded with significantly smaller SiC particles are located in predominance. It was demonstrated that the presence of vitreous carbon in the structure of the formed composite layer lowers and stabilizes the friction coefficient value, but first of all, it reduces the wear of the friction partner in tribological couples. The local reinforcement of castings, especially in the case of using hard ceramic phases, improves the service properties of the products in their areas strongly exposed to wear [26]. At the same time, the process allows maintaining the original mechanical and plastic properties of the alloy matrix in areas free from the reinforcement, which is a favorable feature, especially from the point of view of the composite article finishing process.

The problem that remains to be solved consists of securing the proper quality of the product surface characterized by a high concentration of hard and difficult-to-machine ceramic phase (SiC_p_), i.e., selecting a suitable machining process and optimizing its parameters, especially when maximization of the tool life is taken into account. On the other hand, optimization of the machining process parameters requires a comprehensive approach to understanding the material removal process under the pressure of a cutting tool, as Zhao et al. pointed out [27]. They presented a three-dimensional finite element (3D FE) model for the dynamic material removal process and formation of surface defects in a 50 vol % SiC_p_/5083Al composite manufactured with the use of the powder metallurgy (PM) method. To validate the model experimentally, the single-grit scratch test was used by many authors to evaluate the susceptibility of materials to scratching with a diamond indenter.

The scratch test gives an opportunity to assess the machinability of composite materials, as well as to evaluate their abrasion resistance [28], and it therefore can be used to determine the area and scope of composite product application. At present, a wide variety of devices are used, enabling carrying out such research. Those most interesting, from both application and research points of view, are especially the instruments which, besides providing information about the scratch depth, enable also evaluating the friction force, the coefficient of friction, or the acoustic emission signal which is the effect of degradation of individual microstructure components in the course of scratch formation [29,30,31]. The information, in combination with the results of microstructure examination, allows extending the knowledge of the material degradation mechanism, for instance in the scope of the nucleation and propagation of cracks in microstructure components occurring in the scratch area, and to get to know the characteristic of susceptibility of the material to scratching in greater detail. For example, Kong et al. [32] have discussed the microscratch characteristics and deformation mechanisms of SiC particle-reinforced composites at elevated temperatures. Based on the surface characteristics of the scratching groove, it was indicated that the scratch process mainly includes microcutting, plowing, rubbing, and adhesion.

The objective of the study reported in this paper was to assess the resistance of a centrifugally formed AlSi12/SiC_p_ composite layer with a high share of reinforcing phase (V_p_ > 40%) to scratching with a diamond indenter by performing the scratch test and determining the effect of the load applied to the diamond indenter on the scratch depth and susceptibility of the material to the nucleation and propagation of cracks in hard and brittle SiC particles.

## 2. Materials and Methods 

### 2.1. Matrix and Reinforcement Components 

As the base alloy (composite matrix), AlSi12CuNiMg eutectic silumin was used with the chemical composition shown in Table 1, according to the certificate of the manufacturer (Institute of Non-Ferrous Metals, Light Metals Division in Skawina). Light microscope (LM) and scanning electron microscope (SEM) were used for structural analysis. The microstructure of the Al alloy in its original state is shown in Figure 1. The selection of aluminum alloy with 12 wt % Si content provides a reactivity reduction in the Al/SiC system and decreased potential to forming the brittle carbide phases (i.e., Al_4_C_3_) at the interface boundary. The eutectic network of Al-Si, as well as primary Si particles and different intermetallic phases in particular enriched in magnesium, copper, nickel, iron, and manganese were identified. The presence of these phases correlates with the chemical composition of the alloy. The matrix base alloy was modified by adding 1 wt % Mg in order to improve wettability in the Al/SiC_p_ system [23].

With the intention to improve the service properties of the product, especially its resistance to wear in friction conditions as well as thermal and dimensional stability, hard silicon carbide particles (SiC_p_) marketed by Saint Gobain as F240 grade with the average particle diameter of 50 µm were used as the component reinforcing the plastic aluminum matrix. The surface morphology of silicon carbide particles (SiC_p_—F240) and the particle size distribution histogram are presented in Figure 2. Such kind of reinforcement particle size distribution is advantageous due to the possibility of obtaining a homogeneous composite suspension by the stir casting method [33,34].

### 2.2. Preparation of the Test Material

The reinforcing component in the form of silicon carbide particles (Figure 2) was introduced to modified aluminum alloy using the well-known stir casting method numbered, for many years now, among economically viable liquid-phase PMMCs fabrication methods [33,34,35,36]. The technological process was carried out on a PTA200/PrG station as per an original procedure developed by the present authors and described in detail in previously published papers [10,34,37]. The prepared composite suspension of 10 wt % of SiC_p_ in AlSi12 was cast to graphite forms to obtain a half-product for the subsequent formation of a composite sleeve in the centrifugal casting process. In the next step, composite ingots were melted in a crucible furnace (Figure 3a), which is an integral part of the technological process station MVO500 [38], the design of which enabled tapping the liquid composite suspension directly to a centrifugal mold rotating in the vertical axis system (Figure 3b). 

The selection of proper material parameters such as the reinforcement particles size and type (Figure 2) and matrix alloy chemical composition (Table 1), as well as parameters of the centrifugal casting process in the vertical axis (i.e., the composite suspension pouring temperature as well as the temperature, diameter, and rotational speed of the centrifugal mold, cf. Table 2) enabled making castings in the form of sleeves with an outer composite layer with a thickness of about 6 mm distinctly visible on the cross-section of the casting (Figure 4). Selected model issues concerning the formation of a functional structure in metal–ceramic composites in the centrifugal casting process were already presented in [24].

### 2.3. Microstructure Examination

Specimens for structure examination were cut out from the outer portion of the composite ring with the AlSi12 matrix reinforced with SiC particles. Figure 4c shows locations from which specimens for the examination of structure and the scratch test were taken. The microstructure in the composite layer area was observed on metallographic sections (both unetched and etched). The metallographic sections were etched in 20% NaOH solution for 2 s and for 60 s in the case of deep etching.

The microstructure of the AlSi12/SiC_p_ composite layer characterized by a high content of ceramic reinforcement (V_p_ above 40%) was examined with the use of a GX71 inverted metallurgical microscope (Olympus, Tokyo, Japan) as well as VEGA (TESCAN, Brno, Czech Republic) and S-3400N (Hitachi, Tokyo, Japan) Scanning Electron Microscopes. 

### 2.4. The Scratch Test

The composite layer in the region of high SiC_p_ concentration (V_p_: approximately 40%) was tested for resistance to scratching with the use of the Revetest^®^ Scratch Tester RST (CSM Instruments). Tests were made on the surface of centrifugal casted composite. In all the scratch tests, a Rockwell C-281 type diamond indenter was used with a tip radius of 200 μm and an apex angle of 120°.

To determine the effect of the load force on the scratch depth, constant values of load force on diamond indenters of 2.5 N, 5 N, or 10 N were adopted. The rate of loading the diamond indenter with each of the force values was 5 N/s, whereas the load release rate was 10 N/s. The diamond indenter translation rate was constant and equaled 5 mm/min. The scratch length was 2 mm. The area and scratch direction were marked by a blue arrow in Figure 5.

In the course of the scratch test, values of penetration depth (Pd), friction force (Ff), friction coefficient (FC), and the acoustic emission signal level (AE) were recorded. The value of the acoustic emission index (AE) is quoted in the percentage scale relative to the level of acoustic signal emitted by a standard material, which is titanium nitride (TiN), for which the signal level value of 65 dB is assumed to be 100%.

The microstructure of the area subject to the scratch test was examined with the use of a VEGA Scanning Electron Microscope (TESCAN).

## 3. Results and Discussion

### 3.1. Composite Layer Microstructure

The graded distribution of silicon carbide particles over a cross-section of the tested ring that is characteristic for the centrifugal casting process is shown in Figure 5. The formed 6-mm thick AlSi12/SiC_p_ composite layer is characterized by low porosity (the average void fraction below 1.5%) and a high-volume fraction of reinforcing particles (V_p_ = 40%, especially in the outer region). Examples of the SEM images of the microstructure of the composite layer in the area with high SiC_p_ concentration are shown in Figure 6 and Figure 7. 

As can be seen, the SiC particles in the analyzed outer region of the formed composite layer are uniformly distributed and relatively densely packed in the matrix alloy (Figure 6a and Figure 7a). In addition, the α(Al) + β(Si) eutectics are clearly finer (Figure 6b and Figure 7d) compared to the base alloy shown in Figure 1b. 

Moreover, it can be observed that parts of the eutectic Si phase are heterogeneously nucleated on the SiC particles. It is clearly shown in Figure 6b and Figure 7b, where eutectic Si needles are visible on the ceramic surface reinforcement. The influence of SiC particles on the size and morphology of eutectic silicon in cast A356/SiC_p_ composites have been widely described by Nagarajan et al. [39].

### 3.2. Composite Layer Scratch Testing

Scratch test results enable not only assessing the resistance of the tested material to scratching but also to get to know the mechanism of its degradation occurring as a result of interaction with a diamond indenter. The main intention of the authors was to assess quantitatively the effect of proportionally increasing the interaction force on the degree of the composite material degradation described with parameters such as the penetration depth (Pd), the friction force (Ff), the friction coefficient (FC), and the acoustic emission signal level (AE).

Results of testing the susceptibility of the composite layer with a high volume of SiC particles (V_p_ = 40%) to the formation of a scratch under a diamond indenter loaded with forces of 2.5 N, 5 N, or 10 N are presented in Figure 8, Figure 9 and Figure 10.

In turn, Table 3 summarizes the ranges of variability and average values of penetration depth (Pd), friction force (Ff), friction coefficient (FC), and acoustic emission signal (AE) recorded in the course of making a scratch on the composite specimens under load forces of 2.5 N, 5 N, or 10 N.

The analysis of the scratch test results indicates that with the load force increasing from 2.5 to 10 N, values of the scratch depth, the friction force, and the coefficient friction increase. In addition, an increase in the value of the acoustic emission (AE) signal was observed, the source of which is consecutively nucleating and propagating cracks in SiC particles encountered on the indenter path.

As can be seen, the highest value of the AE signal was found in the case of a scratch made with the load force of 10 N (Figure 10), which indicates that the scratch is characterized by the largest degradation of SiC particles (which can be observed in Figure 13).

The obtained results shown in Table 3 indicate that the proportional increase of value of the load force applied to the diamond indenter causes a quasi-proportional increase of values of the scratch depth (Pd) and the friction force (Ff). A different nature of the recorded changes was observed in case of the friction coefficient (FC) and the acoustic emission (AE) signal level. It can be supposed that in this case, an important role is played by the composite matrix and especially, its plastic properties. The obtained results indicate that with the increasing indentation load, the plasticity of the matrix becomes more and more significant. In the case of small load force values, the degradation of hard reinforcing particles consists of the initiation and propagation of cracks, whereas with an increase of the load, the particles start to crack intensively and are pressed into the plastic matrix. 

The results of SEM observations of the area of the scratch made in the AlSi12/SiC_p_ composite layer by the diamond indenter loaded with forces of 2.5 N, 5 N, or 10 N are presented in Figure 11, Figure 12 and Figure 13, respectively. The effect of interaction with the diamond indenter was an occurrence of cracks in hard SiC particles and smearing of the soft AlSi12 matrix. For that reason, to disclose fully the cracks occurring in SiC particles, an additional deep etching of scratch areas was performed to clean the soft aluminum matrix off particle surfaces. SEM microstructure images after deep etching are shown in Figure 11b,d, Figure 12b,d, as well as Figure 13b,d.

Based on the obtained results, it could be concluded that in case of all the three values of the load force applied to the diamond indenter (2.5 N, 5 N, and 10 N), a similar crack initiation and propagation mechanism occurred. However, a detailed examination of the microstructure in the scratch area shown in Figure 11, Figure 12 and Figure 13 revealed that with increasing load force value, both the penetration depth and scratch width as well as the degree of SiC particles degradation increased. The measures of the degradation are the number of cracks and the degree of development of crack reticulation in the scratch area. The process intensifies with the increasing value of the load force applied to the diamond indenter.

An analysis of the propagation of cracks resulting in the degradation of SiC particles due to operation of the diamond indenter indicates that the process of nucleation and propagation of crack of SiC particles is conditioned by their size, shape (geometry), and orientation relative to the scratching direction. The process is illustrated schematically in Figure 14. In the case of elongated SiC particles characterized with a high length-to-width ratio, the crack propagation direction depends on their orientation relative to the scratch direction. In case of a transversal orientation of the particles, the cracks are parallel to the scratch direction (Figure 14a and Figure 12d), and in case of a longitudinal orientation of the particles, the cracks are perpendicular to the scratching direction (Figure 14b and Figure 12b). That can be explained by an anisotropy of strength properties of the particles due to their geometry. In the case of particles oriented transversely to the scratch direction (Figure 14a), their shear strength is definitely lower in their transversal direction compared to their longitudinal direction. It can be concluded that in that case, shear stresses play the key role in SiC_p_ degradation. 

When SiC particles are oriented lengthwise relative to the scratch direction, the propagation of cracks is due primarily to compressive stresses that are lower in their transversal direction compared to their longitudinal direction.

In scratch areas in which SiC particles are characterized with a lower length-to-width ratio (globular particles) and are oriented perpendicularly (Figure 14c and Figure 11b) or parallel to (Figure 14d and Figure 11d) the scratch direction, the patterns of the cracks’ nucleation and propagation are similar to those shown in Figure 14a,b. However, there is a difference consisting of the main cracks oriented parallel to the scratch direction, as in the case shown in Figure 14c, or the primary cracks oriented perpendicularly to the scratch direction as shown in Figure 14d. Additional cracks occur propagating in the direction perpendicular to the main cracks, and the effect of interaction with the diamond indenter is a development of a reticle of cracks.

In turn, SiC particles situated on the scratch edge (Figure 14e,f and Figure 13b,d) are characterized by the occurrence of cracks oriented perpendicularly to the scratch direction or by a reticle of cracks present only in the area of interaction with the diamond indenter.

The above-described mechanism of the nucleation and development of cracks in SiC particles due to interaction with a diamond indenter is particularly evident in the case of lower load force values (2.5 N and 5 N). For the load force value of 10 N (Figure 13), a significant number of SiC particles was subject to complete disintegration (crushing).

## 4. Conclusions

The obtained test results enable formulating the following conclusions:Susceptibility of the tested AlSi12/SiC_p_ composite layer formed in the centrifugal casting process to scratching depends significantly on the value of force applied to the diamond indenter.With the increasing value of the force applied to the diamond indenter, the friction force and friction coefficient values increase significantly, which is evidence of a high resistance of SiC particles to abrasion.In the process of the diamond indenter forming a scratch in the AlSi12/SiC_p_ composite layer area, the soft matrix is subjected to smearing, while cracks appear in hard and brittle SiC particles.The process of nucleation and propagation of cracks in the course of scratch formation in the AlSi12/SiC_p_ composite layer area is determined by the size and shape of SiC particles and their orientation relative to the scratching direction.Analysis of the acoustic emission signal is a useful tool when the scratch test is used to evaluate the degree of SiC_p_ degradation. With the increasing value of force applied to the diamond indenter, the level of acoustic signal emitted by cracking SiC particles also increases, which provides evidence of the intensification of the material degradation process.

## Figures and Tables

**Figure 1 materials-13-01685-f001:**
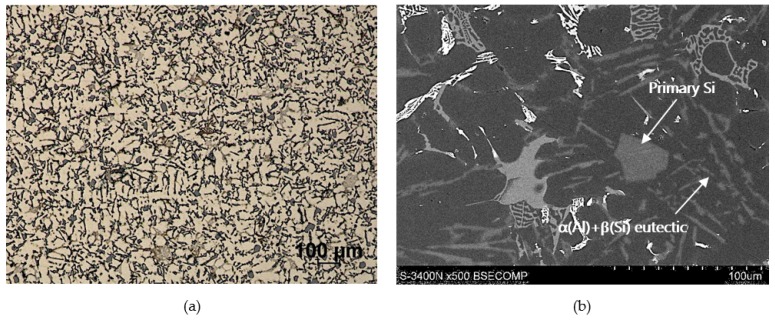
Microstructure of commercial AlSi12CuNiMg (AlSi12) alloy: **(a)** LM; **(b)** SEM.

**Figure 2 materials-13-01685-f002:**
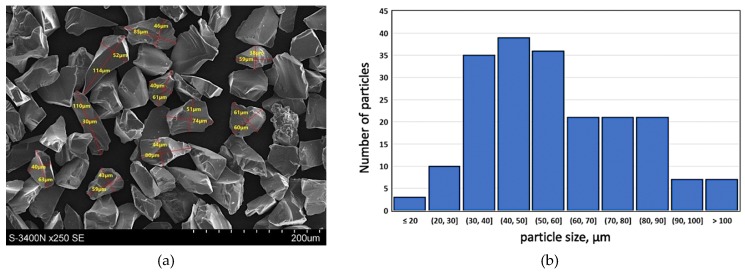
Silicon carbide particles (SiC—F240) used as components reinforcing the aluminum matrix alloy: (**a**) surface morphology, SEM; (**b**) particle size distribution histogram for F240 SiC powder based on performed measurements.

**Figure 3 materials-13-01685-f003:**
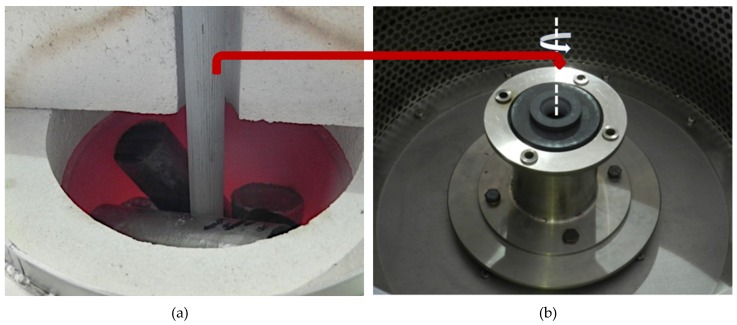
(**a**) A view of the melting chamber of a crucible furnace integrated with a technological process station MVO500; (**b**) a view of centrifugal mold rotating in the vertical axis system.

**Figure 4 materials-13-01685-f004:**
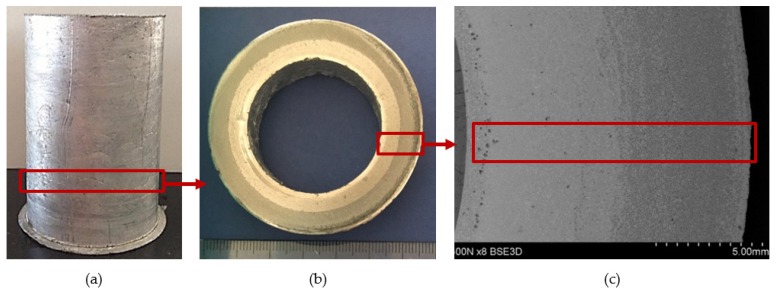
A centrifugal casting of a composite suspension based on AlSi12 alloy with 10% by weight share of SiC_p_: (**a**) a view of the casting in the form of a sleeve; (**b**) a view of ring cut out from the lower portion of the sleeve; (**c**) macrostructure of a ring fragment with reinforcement distribution visible on its cross-section. Unetched section.

**Figure 5 materials-13-01685-f005:**
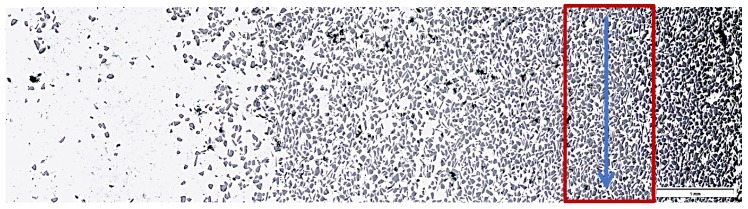
LM microstructure of reinforcing particles distribution on a cross-section of the tested AlSi12/SiC_p_ composite layer; unetched section. The blue arrow indicates the area and direction of the scratch test.

**Figure 6 materials-13-01685-f006:**
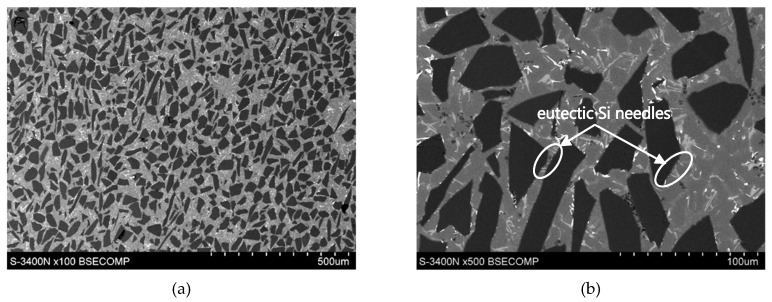
(**a**,**b**) SEM microstructure of the AlSi12/SiC_p_ composite layer in the outer region with the highest amount reinforcement marked in Figure 5; unetched section.

**Figure 7 materials-13-01685-f007:**
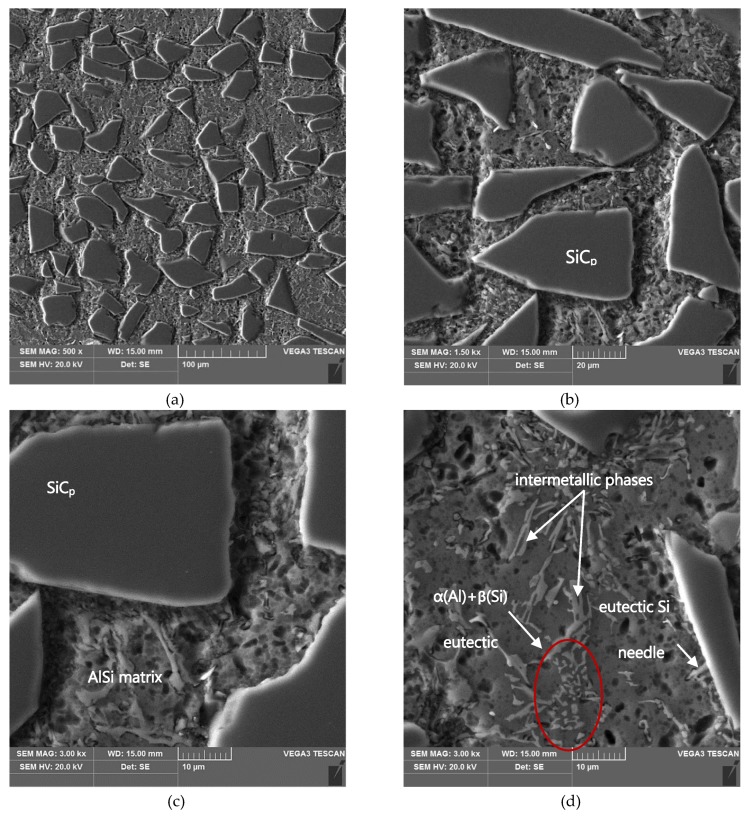
(**a**–**d**) SEM microstructure in the area of formed AlSi12/SiC_p_ composite layer; section etched in 20% NaOH solution for 60 s (deep etching).

**Figure 8 materials-13-01685-f008:**
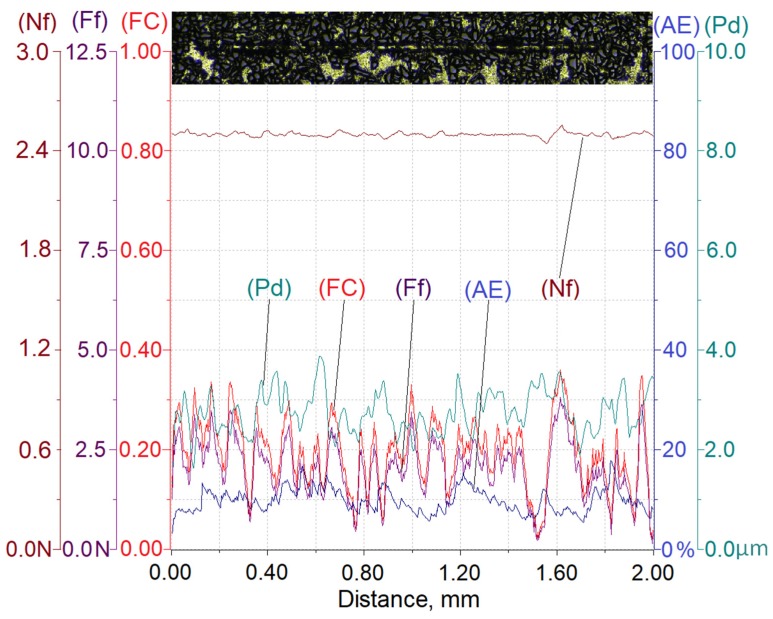
Results of AlSi12/SiC_p_ composite layer scratch testing for the normal load force Nf = 2.5 N; Nf—normal force, Pd—penetration depth, Ff—frictional force, FC—friction coefficient, AE—acoustic emission.

**Figure 9 materials-13-01685-f009:**
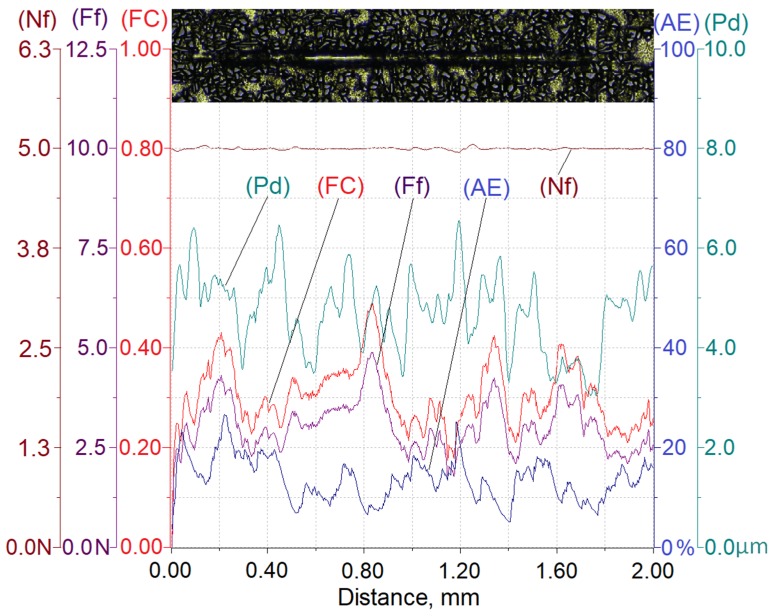
Results of AlSi12/SiC_p_ composite layer scratch testing for the normal load force Nf = 5 N; Nf—normal force, Pd—penetration depth, Ff—frictional force, FC—friction coefficient, AE—acoustic emission.

**Figure 10 materials-13-01685-f010:**
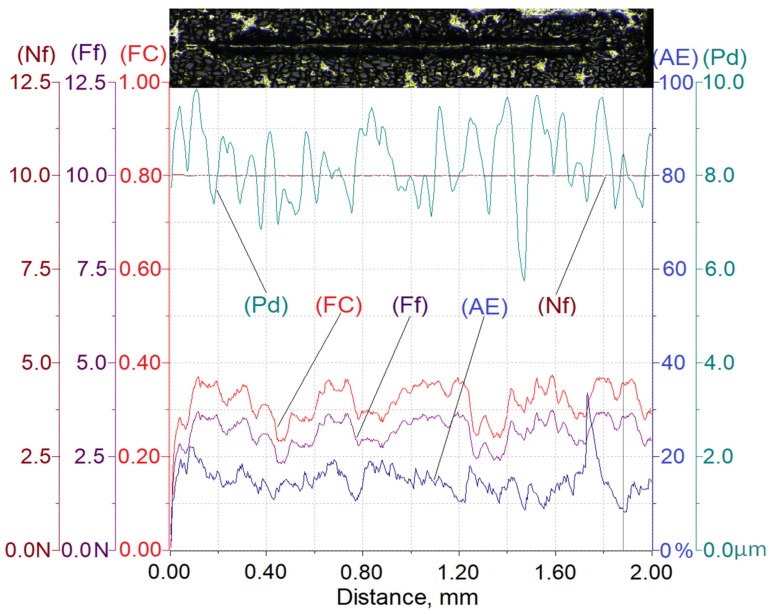
Results of AlSi12/SiC_p_ composite layer scratch testing for the normal load force Nf = 10 N; Nf—normal force, Pd—penetration depth, Ff—frictional force, FC—friction coefficient, AE—acoustic emission.

**Figure 11 materials-13-01685-f011:**
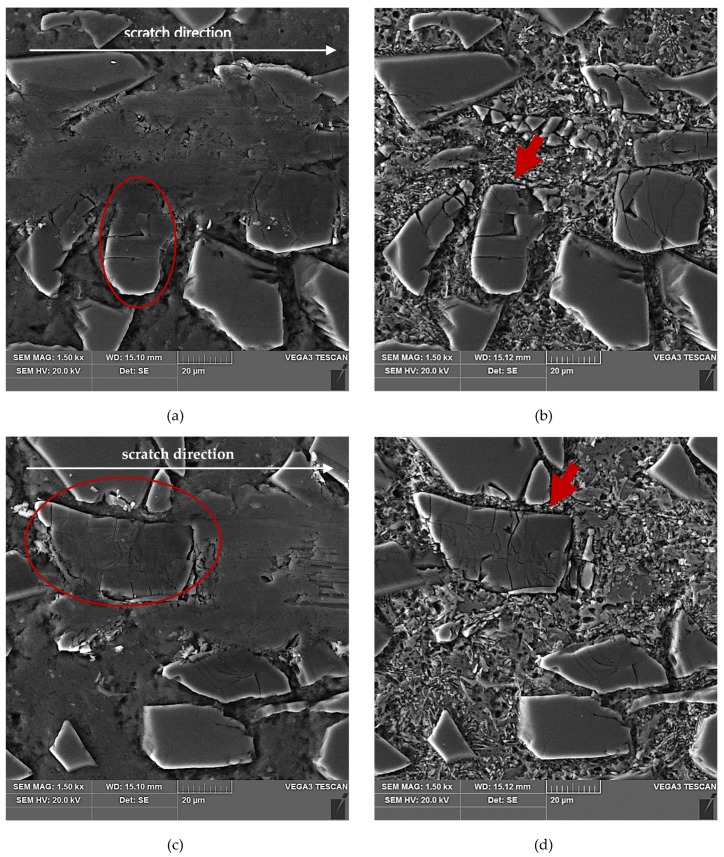
SEM images of a scratch made by the diamond indenter loaded with a force of 2.5 N in the AlSi12/SiC_p_ composite layer area: (**a**) and (**c**) matrix smearing and particles cracking are visible; section etched with 20% NaOH solution for 2 s; (**b**) degradation of the SiC particle oriented perpendicular to the scratch direction has been marked with a red arrow; section etched with 20% NaOH solution for 60 s (deep etching); (**d**) degradation of the SiC particle oriented parallel to the scratch direction has been marked with a red arrow; section etched with 20% NaOH solution for 60 s (deep etching).

**Figure 12 materials-13-01685-f012:**
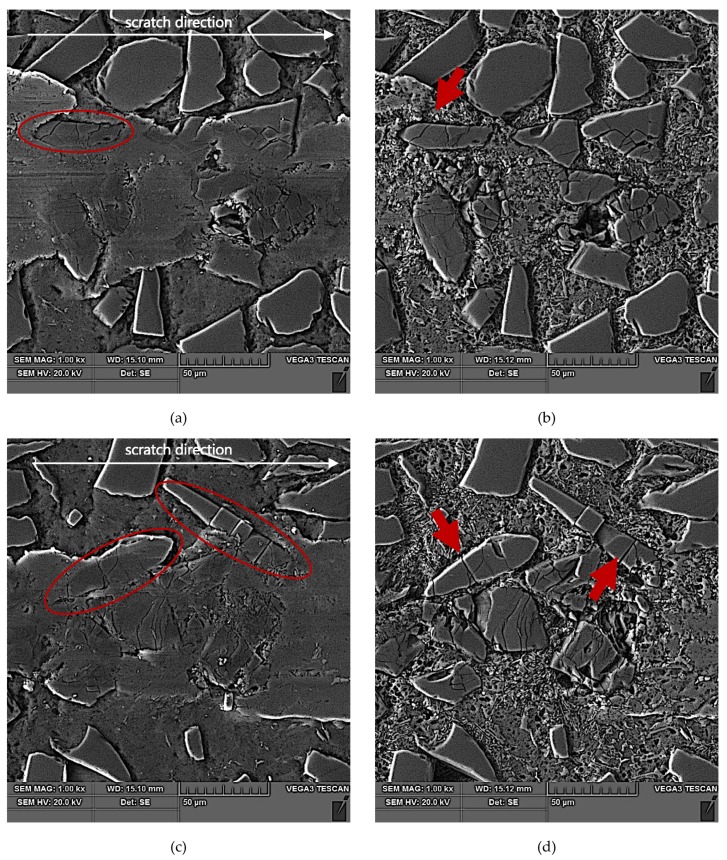
SEM images of a scratch made by the diamond indenter loaded with force of 5 N in the AlSi12/SiC_p_ composite layer area: (**a**) and (**c**) matrix smearing and particles cracking are visible; section etched with 20% NaOH solution for 2 s; (**b**) degradation of the SiC particle oriented longitudinal to the scratch direction has been marked with a red arrow; section etched with 20% NaOH solution for 60 s (deep etching); (**d**) degradation of the elongated SiC particles oriented quasi-parallel to the scratch direction have been marked with a red arrows; section etched with 20% NaOH solution for 60 s (deep etching).

**Figure 13 materials-13-01685-f013:**
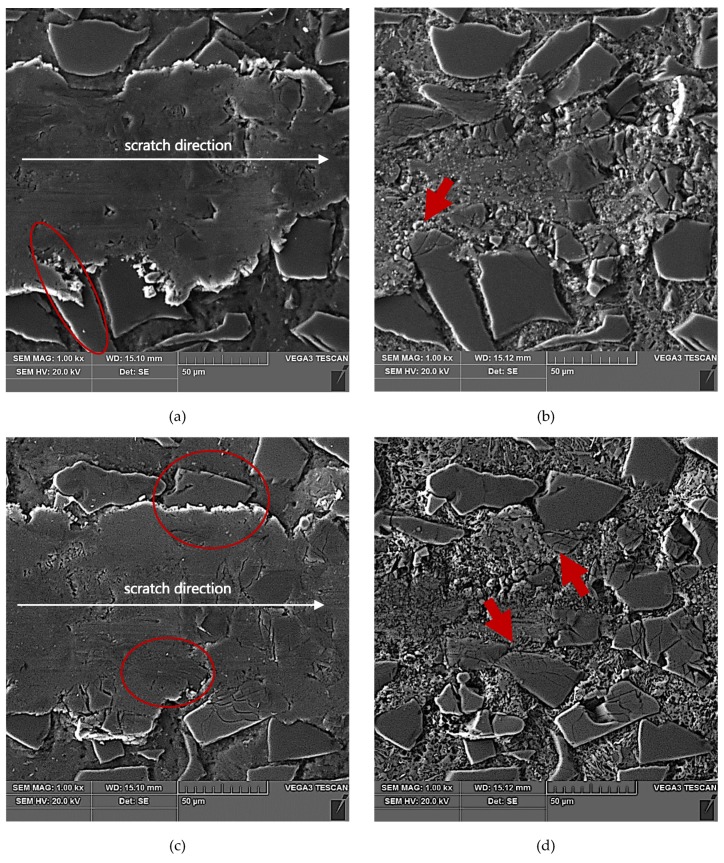
SEM images of a scratch made by the diamond indenter loaded with a force of 10 N in the AlSi12/SiC_p_ composite layer area: (**a**) and (**c**) matrix smearing and particles cracking are visible; section etched with 20% NaOH solution for 2 s; (**b**) and (**d**) degradation of the SiC particles situated on the scratch edge have been marked with red arrows; section etched with 20% NaOH solution for 60 s (deep etching).

**Figure 14 materials-13-01685-f014:**
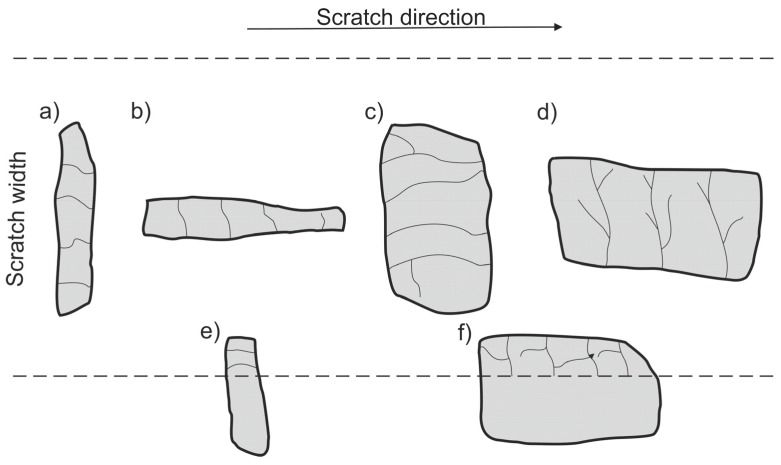
Schematic representation of cracks in SiC particles depending on their size, shape (geometry), and orientation relative to the scratching direction: (**a**) elongated particle with transversal orientation; (**b**) elongated particle with longitudinal orientation; (**c**) globular particle with transversal orientation; (**d**) globular particle with longitudinal orientation; (**e**) elongated particle on the scratch edge; (**f**) globular particle on the scratch edge.

**Table 1 materials-13-01685-t001:** Chemical composition of EN AB-48000 commercial alloy (PN-EN 1676—AlSi12CuNiMg).

	Si	Fe	Cu	Zn	Mn	Ti	Mg	Ni	Al
AlSi12CuNiMg*	11.70	0.43	1.21	0.20	0.18	0.04	1.45	1.19	Bal.

* Alloy composition tested in an IMN OML Skawina accredited laboratory.

**Table 2 materials-13-01685-t002:** Technological parameters of the centrifugal casting process in the vertical axis.

Range of Pouring Temperature (°C)	Centrifugal Mold Temperature (°C)	Centrifugal Mold Diameter (mm)	Centrifugal Mold Rotational Speed (rpm)
720–740	350	60	3000

**Table 3 materials-13-01685-t003:** Values of penetration depth (Pd), friction force (Ff), friction coefficient (FC), and acoustic emission signal (AE) recorded in the course of scratch tests performed in the AlSi12/SiC_p_ composite layer area.

Parameter		Normal Force (N)		
		2.5	5	10
**Pd (μm)**	min.	1.64	3.05	5.76
	max.	3.88	6.55	9.84
	average	2.76	4.8	7.80
**Ff (N)**	min.	0.07	0.67	2.43
	max.	0.91	2.41	3.84
	average	0.49	1.54	3.13
**FC**	min.	0.028	0.135	0.243
	max.	0.364	0.428	0.384
	average	0.196	0.308	0.313
**AE (%)**	min.	5.55	5.20	8.21
	max.	17.85	26.59	33.24
	average	11.70	15.89	20.72

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
