# Peer review of "Scratch Testing of AlSi12/SiCp Composite Layer with High Share of Reinforcing Phase Formed in the Centrifugal Casting Process"

_materials, 2020, doi:10.3390/ma13071685_

Round 1
Reviewer 1 Report
The manuscript ‘Scratch Testing of AlSi12/SiCp Composite Layer with High Share of Reinforcing Phase Formed in the Centrifugal Casting Process' in detail on the resistance of a centrifugally formed AlSi12/SiCp composite layer with high share of reinforcing phase (Vp > 40%) to scratching with a diamond indenter. Some interesting and useful results were obtained. However, I have certain concerns on the following issues.
1: SiC reinforced Al-12Si composite has been widely investigated. Obviously, the addition of SiC can enhance the wear resistance of the composite, however, the mechanical properties would display the opposite trend, how to adjust the percentage of the reinforcement to acquire superior comprehensive properties.
2: In general, the English needs to be improved and the authors are advised to have the manuscript edited by someone proficient in English before resubmission.
3: More references should be added to make comparison with this study.
4: The effects of SiC on the eutectic Si should be discussed.
Author Response
Thank you very much for your valuable comments.
As suggested, the manuscript has been revised and improved (all changes have been highlighted in the text).
Hereby I would like to explain as follows:
Ad. 1. Aluminium alloy matrix composite materials reinforced with high volume of silicon carbide are dedicated particularly for tribological applications. In addition, increase of ceramic particles content in the Al matrix improves both dimensional and thermal stability. In the case of intended applications, the probable decrease on mechanical properties are not so important.
Ad. 2. The proofreading was made by a professional English lecturer and translator.
Ad. 3. As suggested, both the introduction and the references have been extended.
Ad. 4. The presence of ceramic particles during the solidification of the matrix causes refinement both α(Al)+β(Si) eutectic as well as Si eutectic needles. The answer to this remark has been taken into account in improving the text as comment on Figures 6 and 7.
Please find attached a corrected version which containing all changes.

Reviewer 2 Report
The paper deals with the assessment of the ability of the AlSi12/SiCp composite with a high amount of reinforcing phase to resist scratching. The aim of work is interesting and can attract readers and could be beneficial for the journal. But authors should make some changes to improve the quality of the paper and show some novelty.
- The authors should improve the English language.
- Page 3, line 98 – sentence “The scratch test enables ….” is the same as the first sentence in Abstract – there is no need to repeat the information
- Page 3, line 123 – Figure 1. The microstructure of commercial….
- Page 3 – Authors should explain what is in Figure 1, they have to describe microstructures (grains size, presented phases). Not just mentioned that microstructures are in Fig. 1.
- Page 4, line 127 – “…aluminium matrix. The surface….” – there should be the dot not comma.
- Page 4, line 128 – the particle size distribution histogram – authors should summarize the results obtained by analyzing particle size and mention this in the text. What is the result?
- Page 4, line 131 – (b) particle size distribution histogram
- Page 5, line 168 – scanning electron microscopy – capitalize each word, like it is on page 6, line 184
- Page 6, line 189 - it cannot be said that there is a homogeneous distribution of particles, especially if the authors mentioned a region of 6 mm.
- Page 6, line 194 – Figure 5. Microstructure of …
- Page 6, line 196 – from which region pattern for structure analysis was taken? Was it the region with the highest amount of SiCp? The scratch test was made on the surface of centrifugal casted composite or on surface treated in some way?
- Page 7, line 200 – there is no need to show four microstructures of the same material. Especially when there are no differences and higher magnifications do not reveal significant details. Only if the authors want to draw attention to some details. So, do not hesitate and comment on all the figures in the text.
- Authors remain to repeat “… on AlSi12 alloy matrix reinforced with SiCp particles…” such repeating is really not necessary; material did not change during the experiment. Material or experimental material will be also appropriate. (p.6, l.203; p.6 l. 208; p.9, l.224; …)
- Page 9, line 224 - surprised the authors to find that with an increase of load, values of the scratch depth, the friction force, etc. also increased? Did they expect something else? It can be assumed that the value of 10 N was chosen to enhance the effect of the indenter on the material. Why the authors increased the load?
- All experiments (2.5, 5 and 10 N) shown that the same cracks propagation mechanism occurred. Why is necessary to exhibit all, if the results were the same? If authors can compare this material to another material, which behaves differently by the different load it could help to understand the need in three values of the load.
- Are there other works, that deal with the same material or measurement techniques? They should be cited. Authors should compare their own work to other results.
- There is no need for such a number of pictures. For instance, Figure 11(a) and (c) are identical, the only difference is in the scanned position or place of the sample. The same for Figure 11(b) and (d), Figure 12 and 13.
- Page 11, line 245 – „Analysis of propagation of cracks resulting in degradation of SiC particles due to operation of the diamond indenter indicates that the process of nucleation and propagation of SiC particles is conditioned by their size, shape (geometry), and orientation relative to the scratching direction.“ Maybe ...nucleation and propagation of cracks of SiC particles... would be better.
- The observation of SiC particles cracks initiation and proposal of cracks model can contribute to the understanding of the mechanism by which hard particles resist or not against scratching. For better understanding, it will also be beneficial to make visible the grains boundary inside the SiCp.
- The authors should do some discussion about their experimental results. There are just results of experiments, no discussion at all.
In spite of the above comments and on condition that the authors accept the comment, I consider this work to be suitable for publication.
Author Response
Thank you very much for your valuable and relevant comments. All of them were included in the revised manuscript and highlighted in the attached file. Moreover, the proofreading was made by a professional English lecturer and translator. I hope that the changes have improved the readability of the paper.
According to note 19, I kindly explain that this scope of research will be extended and will be the main subject of another paper.
Please find attached a corrected version which containing all changes.

Round 2
Reviewer 1 Report
Accept.